# Effect of Ambient Plasma Treatments on Thermal Conductivity and Fracture Toughness of Boron Nitride Nanosheets/Epoxy Nanocomposites

**DOI:** 10.3390/nano13010138

**Published:** 2022-12-27

**Authors:** Won-Jong Choi, Seul-Yi Lee, Soo-Jin Park

**Affiliations:** Department of Chemistry, Inha University, 100 Inharo, Incheon 22212, Republic of Korea

**Keywords:** polymer-matrix composites (PMCs), interface, thermal conductivity, fracture toughness, surface treatment

## Abstract

With the rapid growth in the miniaturization and integration of modern electronics, the dissipation of heat that would otherwise degrade the device efficiency and lifetime is a continuing challenge. In this respect, boron nitride nanosheets (BNNS) are of significant attraction as fillers for high thermal conductivity nanocomposites due to their high thermal stability, electrical insulation, and relatively high coefficient of thermal conductivity. Herein, the ambient plasma treatment of BNNS (PBNNS) for various treatment times is described for use as a reinforcement in epoxy nanocomposites. The PBNNS-loaded epoxy nanocomposites are successfully manufactured in order to investigate the thermal conductivity and fracture toughness. The results indicate that the PBNNS/epoxy nanocomposites subjected to 7 min plasma treatment exhibit the highest thermal conductivity and fracture toughness, with enhancements of 44 and 110%, respectively, compared to the neat nanocomposites. With these enhancements, the increases in surface free energy and wettability of the PBNNS/epoxy nanocomposites are shown to be attributable to the enhanced interfacial adhesion between the filler and matrix. It is demonstrated that the ambient plasma treatments enable the development of highly dispersed conductive networks in the PBNNS epoxy system.

## 1. Introduction

Boron nitrides (BNs) are identified as an intriguing candidate for fillers in electrically insulating polymer-based nanocomposites due to their relatively high thermal conductivity and good mechanical properties. However, the practical applications of BNs as a reinforcement in epoxy nanocomposites are severely restricted due to the difficulties related to the formation of highly aggregated BN particles during manufacturing and poor interfacial adhesion between the BNs and the epoxy matrix [1,2,3]. In particular, boron nitride nanosheets (BNNS), with a two-dimensional (2D) structure analogous to that of graphene, have been widely utilized as thermal reinforcements due to their high thermal conductivities (400–1000 W∙m^−1^∙K^−1^), outstanding mechanical properties, and chemical stability [4,5,6,7,8,9,10]. Due to their high aspect ratios, the BNNS have advantages for constructing thermal pathways in epoxy nanocomposites, thus enabling improved dispersion and/or reduced mean interparticle distance. However, the ultimate thermal conductivities of the BNNS-loaded epoxy nanocomposites remain far lower than expected due to the strong anisotropy of the BNNS, which leads to a high level of phonon scattering, thus decreasing the interfacial adhesion with the matrix and thereby reducing the thermal conductivity and mechanical properties. Moreover, owing to their high aspect ratios, the BNNS may increase the viscosity of the epoxy nanocomposites at high loading amounts, thus degrading their processability. Moving ahead, the surface functionalization of BNNS may be the most efficient way to overcome these limitations and alleviate the disadvantages of BNNS, thereby leading to an enhancement in the interfacial interactions between the filler and matrix in order to obtain desirable adhesion [11,12,13,14,15,16,17,18,19].

Ambient plasma treatment is a simple, fast, and environmentally benign process because no hazardous chemicals are used. During the treatment process, the physicochemical characteristics of the materials are modified by oxidation, degradation, and cross-linking. In addition, the treatments may cause structural changes within a depth of a few molecular layers while maintaining the inherent properties of the bulk materials. Moreover, plasma treatments have the pragmatic advantage of being performable under an atmosphere, thus ensuring process safety and enabling the fast and effective introduction of oxygen moieties onto the material surface without the formation of unwanted by-products [20,21]. In particular, several studies have reported that plasma treatments can effectively introduce new functionalities onto zirconia (ZrO_2_) surfaces. The surface properties of zirconia modified by these treatments have played an important role in improving the dispersion and enhancing the interfacial adhesion between ZrO_2_ and the epoxy matrix [22,23]. This, in turn, suggests that the plasma treatment of BNNS (PBNNS) would be a promising approach for enhancing the interfacial adhesion in the epoxy nanocomposite system.

Studies have also suggested that such surface enhancements on various fillers may accelerate the conversion of polymer-based nanocomposites into high performance materials. Hence, further research in this direction is expected to provide more rational guidance and fundamental understanding towards the realization of the theoretical limits of interfacial properties [24,25,26,27,28]. This, in turn, could be beneficial in a variety of applications such as medical equipment, heat-releasing materials, and electronic packaging materials. However, to the best of the present authors’ knowledge, there have been no previous studies on the interfacial adhesion between PBNNS and the epoxy matrix, and the related mechanisms have remained unclear. 

The present study is therefore aimed at investigating the effect of PBNNS incorporation upon the interfacial adhesion of conventional epoxy nanocomposites. The chemical properties of PBNNS are investigated using Fourier transform infrared (FT-IR) and X-ray photoelectron spectroscopy (XPS). The structural changes and morphological properties of the PBNNS are observed by X-ray diffraction (XRD) and high-resolution transmission electron microscopy (HR-TEM). Finally, the surface free energy, thermal conductivity and fracture toughness of the PBNNS-loaded epoxy nanocomposites are evaluated.

## 2. Materials and Methods

### 2.1. Materials

The diglycidyl ether of bisphenol-A (DGEBA; YD-128; Kukdo Chemical Co., Seoul, Republic of Korea) was used as the epoxy resin. The epoxide equivalent weight of DGEBA was 185–190 g/equiv., and its density was 1.2 g∙cm^−3^ at 25 °C. Furthermore, 4,4′-diaminodiphenylmethane (DDM, 45–49 g/equiv.; Sigma-Aldrich Co., St. Louis, MO, USA) was used as the curing agent [29]. The chemical structures of DGEBA and DDM is shown in Figure 1. The BNNS (98% purity; Sigma-Aldrich Co., St. Louis, MO, USA) had a lateral size of 150 nm and thickness of 4 nm.

### 2.2. Ambient Plasma Treatment of BNNS

The plasma-treatment system was manufactured by ATMOS-Multi (PLASMART Co., Daejeon, Republic of Korea). Following previous studies, the gases used for the plasma treatments were Ar (99%) and O_2_ (1%). Operating frequency was fixed at 13.56 MHz, the treatment power was 200 W and treatment temperature was 25 °C. The flow rate of Ar/O_2_ mixed gas was 5 L/min, and the plasma treatment speed and the distance between electrodes were 5 mm/s and 7 mm, respectively [30,31,32,33,34]. The plasma-treatment times were varied as 0, 1, 3, 5, 7 and 10 min to generate samples designated hereafter as the pristine BNNS, PBNNS-1, PBNNS-3, PBNNS-5, PBNNS-7, and PBNNS-10.

### 2.3. Fabrication of the PBNNS/Epoxy Nanocomposites

The fabrication process for the PBNNS/epoxy nanocomposites is shown schematically in Figure 2. The PBNNS (30 wt.%) were dispersed in acetone, sonicated for 30 min, then mixed with epoxy resins as in previous studies [35,36,37,38]. The PBNNS/epoxy resin and DDM were then mixed in a planetary mixer for 5 min at room temperature, followed by heating in a vacuum oven at 100 °C for 4 h to evaporate the acetone and remove any bubbles. The mixtures were then cured at 80 °C for 1 h, followed by post-curing at 180 °C for 4 h [39,40]. The specimens of as-received BNNS, the plasma-treated PBNNS, and the PBNNS nanocomposites are designated hereafter as the neat BNNS, the PBNNS, and the PBE nanocomposites, respectively.

### 2.4. Characterization

The structural changes between the neat BNNS and the PBNNS were investigated via X-ray diffraction (XRD; Bruker Co., PHASER, Ettlingen, Germany) under Cu K*α* radiation (λ = 0.154 nm). The surface properties of the PBNNS were investigated using Fourier transform-infrared spectroscopy (FT-IR; Bruker Co., Vertex 80V, Billerica, MA, USA) and X-ray photoelectron spectrometry (XPS; Thermo Fisher Scientific Co., K-Alpha, Waltham, MA, USA). The PBNNS particle size distribution was investigated using dynamic light scattering (DLS) with a zeta-potential analyzer (Microtrac Co., Nanotrac wave ZETA, Montgomeryville, PA, USA). For this measurement, all PBNNS samples were prepared using ethanol under identical colloidal conditions and were ultrasonicated for 1 h. The surface morphologies of the fabricated PBE nanocomposites were investigated using ultra-high resolution scanning electron microscopy (HR-SEM; Hitachi High-Technologies Co., SU8010, Tokyo, Japan) and field emission transmission electron microscopy (FE-TEM; JEOL Co., JEM-2100F, Peabody, MA, USA).

The thermal conductivity of the nanocomposites was determined using a Laser flash analyzer (Netzsch Co, LFA-447, Selb, Germany). Prior to the measurements, both sides of the nanocomposites were coated with graphite spray. The thermal conductivity (*κ*) was then calculated using Equation (1):(1)κ=α ρ Cp
where *α*, *ρ*, and *Cp* are, respectively, the thermal diffusivity, density, and specific heat of the specimen. The specific heat was determined via differential scanning calorimetry (DSC), the density was measured using a density gauge, and the thermal diffusivity was obtained from Equation (2):(2)$$\alpha =\frac{1.38Y^{2}}{\pi^{2}t_{1/2}}\;Cp$$
where *Y* is the thickness of the specimen. The thermal absorption and dissipation capability of the tested nanocomposites were examined using an infrared thermal camera [41,42,43]. For this analysis, the thickness of each sample was controlled at around 2 mm to ensure the same thermal diffusion distance.

The fracture toughness of the nanocomposites was evaluated using a universal testing machine (UTM; Lloyd Instruments Co., LR-5 K plus, Bognor Regis, UK) in accordance with the ASTM D882 standard, and the critical stress intensity factor was determined via a three-point bend test [44]. Here, the notch depth was half the thickness, the crosshead speed was 1 mm∙min^–1^, the specimen size was 0.5 mm × 1 mm × 5 mm, and the span length ratio was 4:1. Under these conditions, the fracture toughness (*K_IC_*) is given by Equation (3):(3)KIC=FLbd32⋅Y
where *F* is the rupture force (N), *L* the span between the supports (mm), *a* the pre-crack length (mm), *b* the specimen width (mm), *d* the specimen thickness (mm), and *Y* a geometrical factor given by Equation (4):(4)$$Y=\frac{3a/d^{1/2}[1.99-\left(a/d\right)\left(1-a/d\right)\left(2.15-3.93a/d+\left(2.7a^{2}/d^{2}\right)\right]}{2\left(1+2a/d\right){\left(1-a/d\right)}^{3/2}}$$

## 3. Results and Discussion

### 3.1. Characterization of the PBNNS

The XRD patterns of the pristine BNNS and the various PBNNS samples are presented in Figure 3a. Here, both spectra exhibit a peak at 2*θ* = 24° due to the (002) reflection plane of the 2-D structure arising from the oxidative degradation of BNNS and PBNNS. Moreover, the diffraction patterns of the PBNNS-1 and PBNNS-7 are similar to that of the pristine BNNS, each exhibiting a diffraction peak at 2*θ* = 45° due to the (100) crystal plane, thereby indicating that these three samples have the same characteristic crystalline structures. However, the XRD pattern of the PBNNS-10 is different from that of the pristine PBNNS, PBNNS-1, and PBNNS-7 samples, thus confirming that the crystalline structure of BNNS is destroyed by excessive plasma-treatment (i.e., more than 10 min), as has been observed previously [45,46,47].

In addition, the FT-IR spectra of the pristine BNNS and the PBNNS are presented in Figure 3b. Here, the pure BNNS exhibits two dominant peaks at 1380 and 800 cm^−1^ due to the B–N stretching vibrations. In addition, the PBNNS samples each exhibit an absorption band at 3360 cm^−1^ due to the stretching vibrations of the hydroxyl (–OH) functional groups introduced by the ambient plasma treatment.

### 3.2. The Surface Morphology of the PBNN

The nanostructures of the pristine BNNS, the PBNNS-7 and the PBNNS-10 are revealed by the TEM images in Figure 4a–c. Here, a comparison of Figure 3a,b clearly confirms the surface modification in the PBNNS-7 due to the etching and cleaning action of the ambient plasma treatment on the BNNS surface. During the ambient plasma treatment, successive oxidation along the sidewalls of the contaminated 2D-layers of the BNNS results in progressive removal of the surface layers. By contrast, Figure 4c reveals the presence of defects and damage to the BNNS structure due to the excessive ambient plasma treatment of the PBNNS-10 sample, thus resulting in the destruction of the 2D-structure [48].

The colloidal stabilities and statistical particle size distributions of the bare Pristine BNNS, PBNNS-7, and PBNNS-10 in distilled water were examined via sedimentation tests and DLS measurements performed 24 h after ultra-sonication for 30 min. In this technique, the random changes of 2D structure materials in the liquid medium are analyzed according to the intensity of scattered light and compared with that of an equivalent hard sphere of ceramic materials. This can provide an estimate of the apparent size of the nanosheet agglomerates in the suspension. A stable colloid will have a constant mean particle size over a period of time, whereas an unstable colloid will exhibit an increase in particle size over time.

The DLS result is shown in Figure 4d indicates that the pristine BNNSs have an average particle diameter of 7–10 μm, which is around 1000 times larger than that of primary particles (5–10 nm). Meanwhile, Figure 4e indicates that PBNNS-1 has an average particle size range of 783–1879 nm with an average of 1331 nm, Figure 4f shows average 12 nm size for PBNNS-7. This could be attributed to the low degree of aggregation in polar solvents due to the presence of hydroxyl functional groups on the surface of BNNS [49,50].

### 3.3. The Interfacial Properties of the PBE Nanocomposites

The surface free energy is an important factor affecting the interfacial adhesion between the PBNNS and the epoxy matrix and, as such, influences the mechanical properties. In the present work, the surface free energy of the nanocomposites was calculated for the liquid and solid forms based on the model of Fowkes [51], Owens [52], and Kaelble [53]. Thus, the total surface free energy (*γ*) is defined as the sum of two components according to Equation (5):(5)$$\gamma =\gamma^{L}+\gamma^{SP}$$
where *γ^L^* is the London dispersion component, and *γ^SP^* is the specific polar component. In addition, the relationship between these two components and the contact angle (CA, or *θ*) is given by Equation (6):(6)$$\gamma^{L}\;\left(1+\cos\theta \right)=2\left(\sqrt{\gamma_{S}^{L}\cdot \gamma_{L}^{L}}+\sqrt{\gamma_{S}^{SP}\cdot \gamma_{L}^{SP}}\right)$$
where $\gamma_{L}^{L}$ and $\gamma_{L}^{SP}$ are the respective components in the liquid phase, $\gamma_{S}^{L}$ and $\gamma_{S}^{SP}$ are the respective components in the solid phase.

The calculated total surface free energies of the neat BNNS and the various nanocomposites are presented in Appendix A and Figure 5a. Here, the neat BNNS exhibits a low surface free energy of 30.9 mJ·m^−2^, which is as expected due to the inert, non-polar surface. By contrast, the PBE nanocomposites exhibit an increase in surface free energy with increasing plasma treatment times. In detail, the surface free energy of the PBE-5 nanocomposite is 41.3 mJ·m^−2^, which represents a significant increase of 34% compared to that of the neat BNNS. Moreover, the highest measured total surface free energy value is 45.1 mJ·m^−2^ for the PBE-7 nanocomposite, with a 46% enhancement relative to the neat BNNS. In particular, the specific polar components of the PBE nanocomposites are found to exert a greater influence on the surface free energy than the London dispersion components. In view of the above-mentioned FT-IR and XPS results, this could be attributed to an increase in the number of hydroxyl functional groups on the PBNNS surfaces provided by the ambient plasma treatment. Consistently, the results indicate a decrease in the contact angle with increased plasma-treatment time, which can also be attributed to the increased number of hydroxyl groups from which distilled water droplets can rapidly diffuse, thus providing an improved wettability for the nanocomposites.

### 3.4. The Thermal Conductivity of the PBE Nanocomposites

The effects of plasma-treatment time upon the thermal conductivity of the PBE nanocomposites are indicated in Figure 6a and Appendix A. Here, the thermal conductivity is seen to increase with increasing treatment time up to 7 min (PBE-7) and to decrease thereafter, as expected. In detail, the thermal conductivity of the PBE-5 nanocomposite is 0.446 W∙m∙K^−1^, which represents a significant increase of 20% compared to that of the neat BNNS (0.372 W∙m^−1^∙K^−1^). Moreover, the highest measured thermal conductivity is 0.535 W∙m∙K^−1^ for the PBE-7 nanocomposite, with a 44% enhancement compared to the neat BNNS. Furthermore, the plot in Figure 6b evidently shows a linear relationship between specific polar components and thermal conductivity, thus demonstrating the potential of PBNNS as a thermal interface material that can effectively promote thermal conductivity in nanocomposites [7,54,55]. Thus, the thermal conductivity of the nanocomposites is influenced not only by the type of filler but also by the interfacial interaction between the filler and the matrix. In detail, the PBE nanocomposites exhibit behavior in which increasing specific polar components of the surface free energy promote the formation of thermal conductivity pathways. These results are consistent with previous studies indicating that enhanced interfacial interaction and adhesion are important for increasing thermal conductivity, in agreement with previous studies [56,57,58]. The proposed thermal network mechanism of thermal conductivity is shown schematically in Figure 6c. For the neat BNNS, there is the possibility for aggregate formation, as demonstrated by the above-mentioned DLS and surface free energy analyses. Therefore, it is difficult to have a thermal conductivity mechanism within the epoxy matrix due to the poor dispersion and interfacial adhesion. By contrast, in the case of the PBE nanocomposites, the enhanced dispersion and interfacial adhesion in the absence of aggregation leads to the formation of thermally conductive networks and, hence enhanced thermal conductivity [59,60].

To evaluate the effect of PBNNS composites for thermal conductivity performances, a comparison between the PBE and other BN composites. As shown in Figure 6b and Table 1, PBE achieve far superior thermal conductivity performance (0.535 W∙m^−1^∙K^−1^) compared to all other BN composites [61,62,63,64,65].

The change in heat absorption of the neat BNNS and the various PBE nanocomposites according to plasma-treatment time is indicated by the plots of temperature against heating time in Figure 7a and the corresponding infrared thermal camera images in Figure 7c–d. Here, the surface temperature of each sample is seen to increase with heating time, but the PBE-7 nanocomposite (Figure 7b) exhibits the highest temperature compared with the other samples over the same time period. Similarly, during the cooling process, the surface temperature of the PBE-7 nanocomposite decreases more rapidly than that of the other samples, reaching a minimum value and remaining stable thereafter. The faster heating and cooling rates of this sample indicate its higher thermal conductivity and rate of thermal diffusion [66].

### 3.5. Fracture Toughness of PBE Nanocomposites

The relationship between the specific polar components of surface free energy and the fracture toughness (*K_IC_*) of the samples is indicated in Figure 8a, with the PBE nanocomposites exhibiting good linearity. These results indicate that the increase in interfacial adhesion is important for enhancing the fracture toughness of the PBE nanocomposites. The highest measured thermal conductivity value is 21.4 MPa.m^1/2^ for the PBE-7 nanocomposite, indicating a 110% enhancement compared to the neat BNNS (10.7 MPa.m^1/2^). As noted above, previous studies have demonstrated that the nanocomposites fabricated with PBNNS generally yield higher *K_IC_* values than other types of nanocomposites [67,68]. This increase in the *K_IC_* is believed to be related to the non-aggregated, well-distributed, and tightly embedded PBNNS within the epoxy matrix, as revealed by the fractured surfaces of the various nanocomposites in Figure 8c–e. This means that any cracks formed will propagate through the epoxy matrix and around the PBNNS due to the outstanding dispersion and interfacial adhesion between the PBNNS and the epoxy matrix, thus leading to good fracture and pull-out resistance [69,70]. Thus, the SEM image of the neat BNNS in Figure 8c shows typically catastrophic fractures due to aggregated BNNS and poor interfacial adhesion within the epoxy matrix. By contrast, the PBE-7 nanocomposite (Figure 8d) exhibits a rough fracture surface due to the lack of aggregation and the resulting enhancement in the dispersion and interfacial adhesion with the epoxy matrix. However, the PBNNS-10 (Figure 8e) exhibits similar levels of aggregation to that of the neat nanocomposite, thus indicating decreased dispersion and interfacial adhesion due to the excessive plasma-treatment time. These observations confirm the above-mentioned conclusion that the optimum plasma-treatment time is 7 min (PBNNS-7). It is important to note that the selected plasma-treatment times were within a reasonable range, that the optimal specific polar component was derived therefrom, and that the SEM images of the fracture surface confirmed that improved fracture toughness could be obtained [71].

## 4. Conclusions

The effects of the plasma-functionalization of BNNS upon the thermal conductivity and fracture toughness of epoxy nanocomposites were investigated herein. In particular, the effects of varying the treatment time (1, 3, 5, 7, and 10 min) were examined. In addition, dynamic light scattering (DLS) and high-resolution scanning electron microscopy (HR-SEM) analyses were used to further examine the effects of plasma-functionalization upon the dispersion state and interfacial adhesion of the BNNS by observing the fracture surfaces. In summary, substantial enhancements in the thermal conductivity and fracture toughness of the BNNS/epoxy (PBE) nanocomposites were achieved by using the optimal plasma-treatment time of 7 min. The resulting PBE-7 nanocomposite exhibited a higher specific polar component of the surface free energy than the neat BNSS, and the attached hydroxyl functional group effectively inhibited the BNNS aggregation within the epoxy matrix. In addition, the PBE-7 nanocomposite exhibited a 44% higher thermal conductivity and a 110% higher fracture toughness than the neat BNNS due to its high dispersion and strong interfacial adhesion. The results of this study explicated that a strong correlation exists between the surface free energy and the thermal conductivity and fracture toughness of nanocomposites.

## Figures and Tables

**Figure 1 nanomaterials-13-00138-f001:**
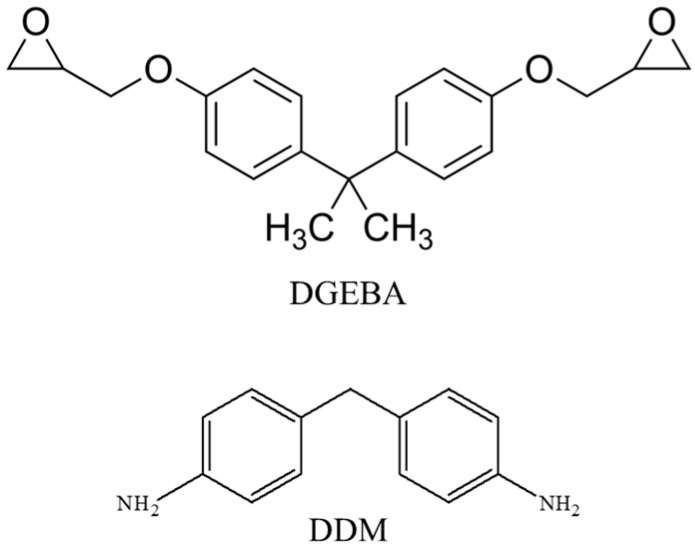
Chemical structures of DGEBA and DDM.

**Figure 2 nanomaterials-13-00138-f002:**
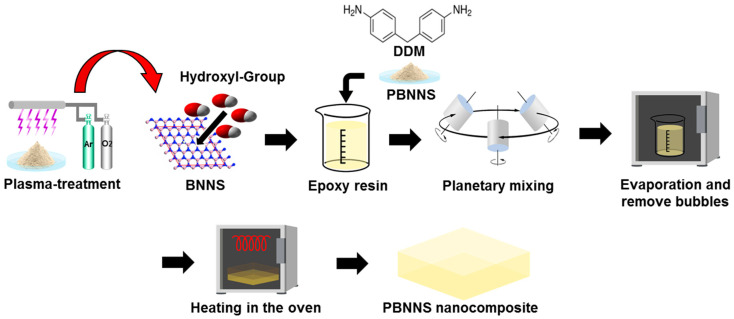
Schematic representation of preparation process on PBNNS nanocomposites.

**Figure 3 nanomaterials-13-00138-f003:**
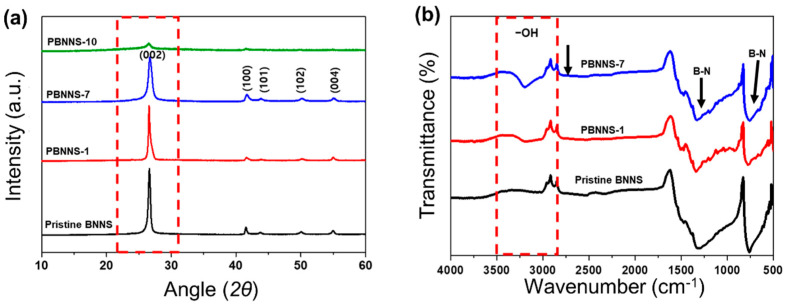
Characterization of PBNNS: (**a**) XRD spectra and (**b**) FT-IR spectra.

**Figure 4 nanomaterials-13-00138-f004:**
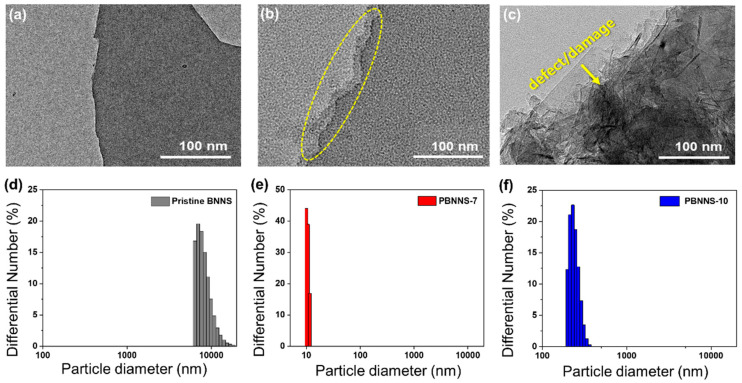
Surface morphology and particle distribution of PBNNS (**a**) TEM images of pristine BNNS, (**b**) PBNNS-7, (**c**) PBNNS-10, particle-size distributions of (**d**) pristine BNNS, (**e**) PBNNS-7, and (**f**) PBNNS-10.

**Figure 5 nanomaterials-13-00138-f005:**
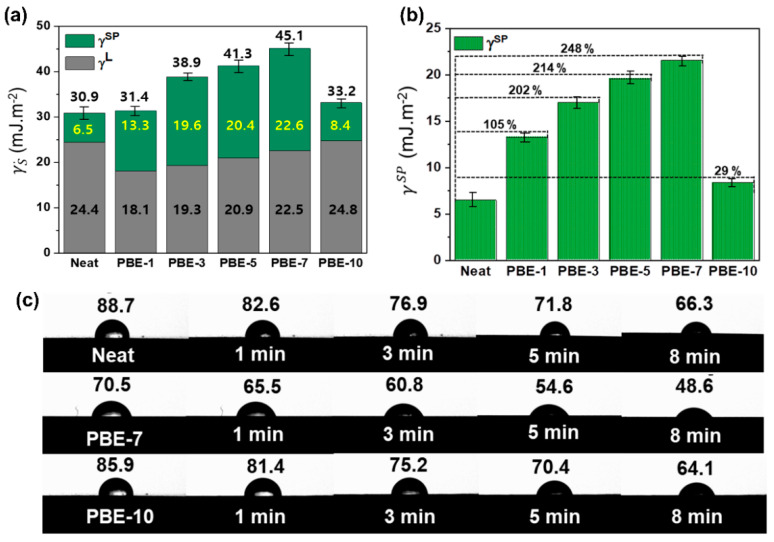
Interfacial properties of the BNNS/epoxy nanocomposites: (**a**) surface free energy, (**b**) specific polar component, and (**c**) optical images of the contact angles of distilled water over time.

**Figure 6 nanomaterials-13-00138-f006:**
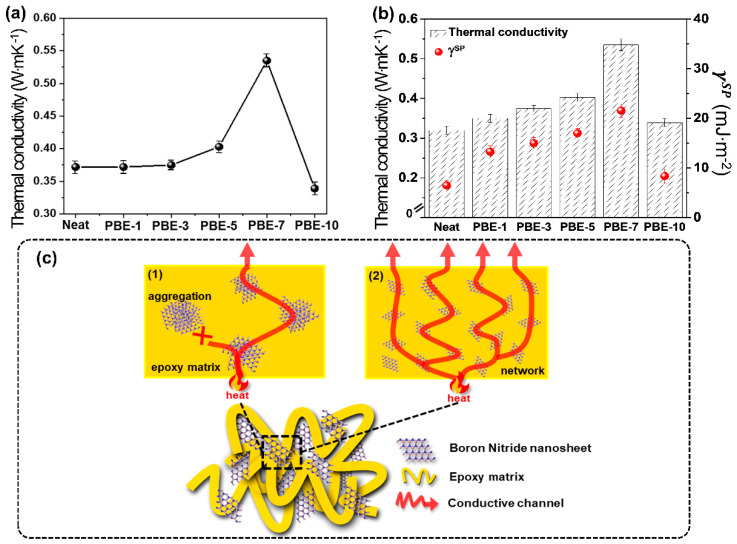
Thermal conductivity of the BNNS/epoxy nanocomposites: (**a**) thermal conductivity, (**b**) correlation between specific polar component of surface free energy and thermal conductivity, (**c**) a schematic representation of thermal conductivity mechanisms of Neat and PBE-7.

**Figure 7 nanomaterials-13-00138-f007:**
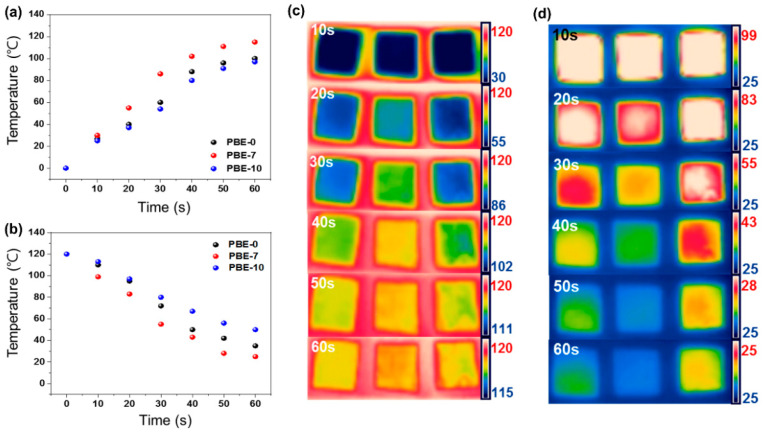
Infrared thermal camera of the BNNS/epoxy nanocomposites: (**a**) temperature change curve of heat absorption over time, (**b**) temperature change curve of heat dissipation over time, (**c**) optical images of the heat absorption over time, and (**d**) optical images of the heat dissipation time.

**Figure 8 nanomaterials-13-00138-f008:**
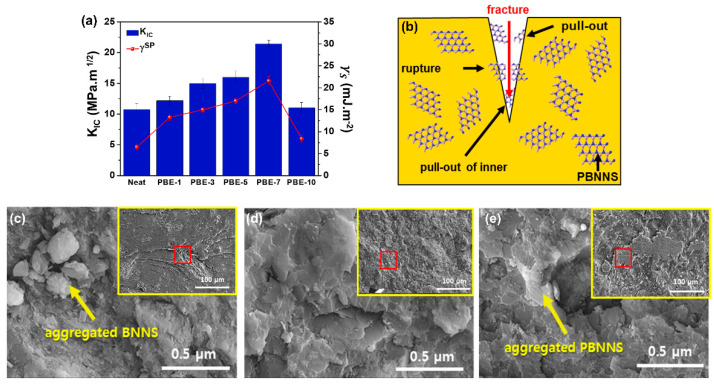
Fracture toughness of the BNNS/epoxy nanocomposites: (**a**) Correlation between specific polar component of surface free energy and *K_IC_*, (**b**) fracture mechanism, (**c**) fracture surfaces of the Neat, (**d**) PBE-7, and (**e**) PBE-10.

**Table 1 nanomaterials-13-00138-t001:** Comparison of various BN/epoxy composites.

Composite	Filler Contents (wt.%)	Thermal Conductivity (W∙m^−1^∙K^−1^)	Ref.
PBE-7	30.0	0.535	This work
BN/EP	30.0	0.382	[61]
KH550 modified BN	30.0	0.390	[61]
nano-h-BN	40.0	0.478	[62]
Functionalized BN	38.0	0.450	[63]
Epoxy/h-BN composite	34.4	0.321	[64]
Epoxy	–	0.054	[65]

## Data Availability

The data presented in this study are available on request from the corresponding author.

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
