# Peer review of "Effect of Ambient Plasma Treatments on Thermal Conductivity and Fracture Toughness of Boron Nitride Nanosheets/Epoxy Nanocomposites"

_nanomaterials, 2022, doi:10.3390/nano13010138_

Round 1
Reviewer 1 Report
The authors investigated the effects of the plasma-functionalization of BNNS on the thermal conductivity and fracture toughness of epoxy nanocomposites. Generally, the results are meaningful and reasonable. However, authors should address issues as follow before publication:
1. Please introduce the structure of plasma-treatment system in Fig.1, how to ensure BNNS particles be treated evenly?
2. In the preparation process of nanocomposites, why add the DDM?
3. In the DLS results, why the mean particle diameter of the pristine BNNS is larger than that of the primary particles? What’s the primary particles?
4. The authors mentioned hydroxyl functional group effectively inhibited the BNNS aggregation. How to evaluate the aggregation degree? Zeta potential is related to dispersity and aggregation closely, it is suggested to measure Zeta potential.
5. In Fig. 4(c), how many times measured the contact angle? How about the Repeatability?
6. It is mentioned that“Consistently, the results indicate a decrease in the contact angle with increased plasma-treatment time, which can also be attributed to the increased number of hydroxyl groups from which distilled water droplets can rapidly diffuse, thus providing an improved wettability for the nanocomposites”. However, this conclusion cannot be obtained from Fig. 4(c). Compared with neat's results, the contact angles of distilled water of PBE-7 decreases, but that of PBE-10 increases.
7. Is the thermal conductivity from the multiple measurements under the same conditions in Fig 5(a)? It is better to add error bar.
8. In Fig. 6(a) and (b), the endothermic temperature of PBE-7 reaches 120℃, while that of PBE-0 and PBE-10 does not reach 120℃. Please explain why the initial temperature of heat dissipation is 120℃.
9. In this paper, the filling ratio is 30 wt.%. Is it possible to increase the filling ratio after plasma treatment?
10. Actually, there are many research about the treatment of BNNS, it is suggested to compare the treatment effect with other research.
Author Response
Response to the Referee’s Comments
|
“Effect of ambient plasma treatments on thermal conductivity and fracture toughness of boron nitride nanosheets/epoxy nanocomposites”
The general revisions of the manuscript are noted in the manuscript under the reviewer’s direction.
[Reviewer #1]
- Note of Revised Version:
- Please introduce the structure of plasma-treatment system in Fig.1, how to ensure BNNS particles be treated evenly?
à We really appreciated that you took time to review my paper and gave us the valuable comments towards enhancing the quality of this manuscript. We were additionally mentioned about introduce the structure of plasma-treatment system on this paper in introduction parts. Thanks for your revise.
<Revised or added paragraph and figure>
2.2 Ambient plasma treatment of BNNS: Page 2, Line 89-96;
The plasma-treatment system was manufactured by ATMOS-Multi (PLASMART co., Korea). Following previous studies, the gases used for the plasma treatments were Ar (99%) and O2 (1%). Operating frequency was fixed at 13.56 MHz, the treatment power was 200 W and treatment temperature was 25 ℃. The flow rate of Ar/O2 mixed gas was 5 l/min, and the plasma treatment speed and the distance between electrodes were 5 mm/s and 7 mm, respectively. The plasma-treatment times were varied as 0, 1, 3, 5, 7 and 10 min to generate samples designated hereafter as the pristine BNNS, PBNNS-1, PBNNS-3, PBNNS-5, PBNNS-7, and PBNNS-10.
- In the preparation process of nanocomposites, why add the DDM?
à Thanks for your valuable reviews for enhancing the quality of our paper. We added the DDM for the epoxy resin hardener in the preparation process. Amine type curing agents are one of the basic curing agents for epoxy resins, and they can be classified into three major categories: aliphatic, aromatic, or cycloaliphatic amines. Amine type curing agents react with epoxide rings by nucleophilic addition. Fig. 16 shows the chemical structures of 4,4′-diaminodiphenyl methane (DDM) and 4,4′-diaminodiphenyl sulfone (DDS). The figure shows the cure reaction mechanism of amine and epoxide. Thanks for your precious time and valuable comments again.
<Revised or added paragraph and figure>
2.1. Materials: Page 2, Line 80-83;
The diglycidyl ether of bisphenol-A (DGEBA; YD-128; Kukdo Chemical Co., Korea) was used as the epoxy resin. The epoxide equivalent weight of DGEBA was 185-190 g/equiv., and its density was 1.2 g∙cm-3 at 25 °C. Furthermore, 4,4'-diaminodiphenylmethane (DDM, 45–49 g/equiv.; Sigma-Aldrich Co., USA) was used as the curing agent [29].
Fig. 1. Chemical structures of DGEBA and DDM.
[29] Jin, F. L., Li, X., & Park, S. J. (2015). Synthesis and application of epoxy resins: A review. Journal of Industrial and Engineering Chemistry, 29, 1-11.
- In the DLS results, why the mean particle diameter of the pristine BNNS is larger than that of the primary particles? What’s the primary particles?
à Thanks for your valuable time and contribution to our work. We have carefully examined and revised the manuscript in a thorough manner according to the reviewer’s suggestion. A major problem associated with the use of BNSS in epoxy-based nanocomposite is their high hydrophobicity and extremely low dispersibility, which can lead to irreversible agglomeration in a short period of time due to van der Waals interactions between the prismatic surfaces. Thanks for your kind review.
<Revised or added paragraph and figure>
3.2. The surface morphology of the PBNN: Page 5, Line 173-181;
The colloidal stabilities and statistical particle size distributions of the bare Pristine BNNS, PBNNS-7, and PBNNS-10 in distilled water were examined via sedimentation tests and dynamic laser scattering (DLS) measurements performed 24 h after ultra-sonication for 30 min. In this technique, the random changes of 2D structure materials in the liquid medium are analyzed according to the intensity of scattered light and compared with that of an equivalent hard sphere of ceramic materials. This can provide an estimate of the apparent size of the nanosheet agglomerates in the suspension. A stable colloid will have a constant mean particle size over a period of time, whereas an unstable colloid will exhibit an increase in particle size over time.
- The authors mentioned hydroxyl functional group effectively inhibited the BNNS aggregation. How to evaluate the aggregation degree? Zeta potential is related to dispersity and aggregation closely, it is suggested to measure Zeta potential.
à Thanks for your kind comment. Unfortunately, we are not able to provide the Zeta potential at this moment (We have requested from several places, but Zeta potential measurement is limited). But also, we could derive the BNNS aggregation through the DLS results. Thanks for your precious time and valuable comments again.
- In Fig. 4(c), how many times measured the contact angle? How about the Repeatability?
à Thanks for your questions. We measured the contact angle 5 times each sample. The average error range of contact angle is Ɵ. Thanks for your review.
Table. The contact angles of PBE nanocomposites
|
Specimens |
Contact angle (θ) |
||
|
Distilled water |
Diiodomethane |
Ethylene glycol |
|
|
Neat |
88.7 ±0.3 |
59.1 ±0.9 |
38.1 ±0.7 |
|
PBE-1 |
84.5 ±0.4 |
58.0 ±0.4 |
39.5 ±0.1 |
|
PBE-3 |
80.8 ±0.1 |
57.1 ±0.9 |
38.3 ±0.5 |
|
PBE-5 |
75.9 ±0.2 |
56.8 ±0.9 |
39.8 ±0.8 |
|
PBE-7 |
70.5 ±0.6 |
56.1 ±0.6 |
38.0 ±0.9 |
|
PBE-10 |
85.9 ±0.4 |
58.8 ±0.1 |
38.2 ±0.7 |
- It is mentioned that “Consistently, the results indicate a decrease in the contact angle with increased plasma-treatment time, which can also be attributed to the increased number of hydroxyl groups from which distilled water droplets can rapidly diffuse, thus providing an improved wettability for the nanocomposites”. However, this conclusion cannot be obtained from Fig. 4(c). Compared with neat's results, the contact angles of distilled water of PBE-7 decreases, but that of PBE-10 increases.
à Thanks for your kind comment. Contact angle measurements (CA) are used to determine the surface free energy and evaluate the interfacial properties of nanocomposites. A higher surface free energy leads to better wettability and thus enhances the interfacial properties of the resulting nanocomposites. In this study, the surface free energy of the PBE-10 nanocomposite is 33.2 mJ.m−2, which represents considerable increases of ~7.4% compared to that of Neat nanocomposites (30.9 mJ.m−2). Moreover, the polar component for the PBE-10 nanocomposites was found to be 8.4 mJ.m−2, equating to enhancements of ~ 29%, compared to that of Neat nanocomposites (6.5 mJ.m−2) at the same filler content. These findings are further supported by wetting behaviors of the nanocomposites. The wetting behavior of the Neat nanocomposite is kept at approximately 66.3°, moreover the wetting behavior of the PBE-10 nanocomposites approximately 64.1°. In the context of the previously mentioned FT-IR and XPS study results, we found that P-BNNS gave an increase in hydroxyl functional groups supplied by plasma treatment, which could have been the main contribution toward the enhanced polar components. This decrease is due to the enhanced wettability, resulting from a hydroxyl group attached to the BNNS surface. Thanks for your kind review.
- Is the thermal conductivity from the multiple measurements under the same conditions in Fig 5(a)? It is better to add error bar.
à Thanks for your very professional advice. Added error bar to Fig. 5(a). Thanks for your precious time and valuable comments again.
- In Fig. 6(a) and (b), the endothermic temperature of PBE-7 reaches 120℃, while that of PBE-0 and PBE-10 does not reach 120℃. Please explain why the initial temperature of heat dissipation is 120℃.
à Thanks for your very professional advice. Infrared thermal camera measurement is one of the most suitable instruments for determining the temperature of heat dissipation of epoxy nanocomposite laminates. Therefore, the measurement temperature 120 ℃ has been widely used in various professional studies [1], [2], [3]. Thank you for your thoughtful comment.
[1] Shim, H. B., Seo, M. K., & Park, S. J. (2002). Thermal conductivity and mechanical properties of various cross-section types carbon fiber-reinforced composites. Journal of materials science, 37(9), 1881-1885.
[2] Zhang, Y., & Park, S. J. (2019). Imidazolium-optimized conductive interfaces in multilayer graphene nanoplatelet/epoxy composites for thermal management applications and electroactive devices. Polymer, 168, 53-60.
[3] Kim, S. H., Rhee, K. Y., & Park, S. J. (2020). Amine-terminated chain-grafted nanodiamond/epoxy nanocomposites as interfacial materials: Thermal conductivity and fracture resistance. Composites Part B: Engineering, 192, 107983.
- In this paper, the filling ratio is 30 wt.%. Is it possible to increase the filling ratio after plasma treatment?
à Thanks for your questions. Adding 30 ~ 40 wt.% of BNNS filler to epoxy nanocomposites has been extensively studied. We referenced widely used in various professional studies [1], [2], [3], [4]. Thanks for your kind review.
[1] Liu, Z., Li, J., Zhou, C., & Zhu, W. (2018). A molecular dynamics study on thermal and rheological properties of BNNS-epoxy nanocomposites. International Journal of Heat and Mass Transfer, 126, 353-362.
[2] Zhao, L., Yan, L., Wei, C., Li, Q., Huang, X., Wang, Z., ... & Ren, J. (2020). Synergistic enhanced thermal conductivity of epoxy composites with boron nitride nanosheets and microspheres. The Journal of Physical Chemistry C, 124(23), 12723-12733.
[3] Xiao, C., Guo, Y., Tang, Y., Ding, J., Zhang, X., Zheng, K., & Tian, X. (2020). Epoxy composite with significantly improved thermal conductivity by constructing a vertically aligned three-dimensional network of silicon carbide nanowires/boron nitride nanosheets. Composites Part B: Engineering, 187, 107855.
[4] Wu, N., Yang, W., Li, H., Che, S., Gao, C., Jiang, B., ... & Li, Y. (2022). Amino acid functionalized boron nitride nanosheets towards enhanced thermal and mechanical performance of epoxy composite. Journal of Colloid and Interface Science, 619, 388-398.
- Actually, there are many research about the treatment of BNNS, it is suggested to compare the treatment effect with other research.
à Thanks for your very professional advices. We were additionally mentioned about superiority of this work in thermal conductivity section. Thanks for your revise.
3.4. The thermal conductivity of the PBE nanocomposites: Page 8, Line 252-255;
To evaluate the effect of PBNNS composites for thermal conductivity performances, a comparison between the PBE and other BN composites. As shown in Fig. 5(b) and Table 1, PBE achieve far superior thermal conductivity performance (0.535 W∙m-1∙K-1) compared to all other BN composites.
Table. 1. Comparison of various BN/epoxy compoistes
|
Composite |
Filler contents (wt.%) |
Thermal conductivity (W∙m-1∙K-1) |
Ref. |
|
PBE-7 |
30.0 |
0.535 |
This work |
|
BN/EP |
30.0 |
0.382 |
[59] |
|
KH550 modified BN |
30.0 |
0.390 |
[59] |
|
nano-h-BN |
40.0 |
0.478 |
[60] |
|
Functionalized BN |
38.0 |
0.450 |
[61] |
|
Epoxy/h-BN composite |
34.4 |
0.321 |
[62] |
|
Epoxy |
– |
0.054 |
[63] |
[59] Gu, J., Zhang, Q., Dang, J., & Xie, C. (2012). Thermal conductivity epoxy resin composites filled with boron nitride. Polymers for Advanced Technologies, 23(6), 1025-1028.
[60] Yang, H., Chen, Q., Wang, X., Chi, M., Liu, H., & Ning, X. (2019). Dielectric and thermal conductivity of epoxy resin impregnated nano-h-BN modified insulating paper. Polymers, 11(8), 1359.
[61] Teng, C. C., Ma, C. C. M., Chiou, K. C., Lee, T. M., & Shih, Y. F. (2011). Synergetic effect of hybrid boron nitride and multi-walled carbon nanotubes on the thermal conductivity of epoxy composites. Materials Chemistry and Physics, 126(3), 722-728.
[62] Wang, Z., Iizuka, T., Kozako, M., Ohki, Y., & Tanaka, T. (2011). Development of epoxy/BN composites with high thermal conductivity and sufficient dielectric breakdown strength partI-sample preparations and thermal conductivity. IEEE Transactions on Dielectrics and Electrical Insulation, 18(6), 1963-1972.
[63] Garrett, K. W., & Rosenberg, H. M. (1974). The thermal conductivity of epoxy-resin/powder composite materials. Journal of Physics D: Applied Physics, 7(9), 1247.
We really appreciated that you took time to review my paper and gave us the valuable comments towards enhancing the quality of this manuscript.
Sincerely yours,
Soo-Jin Park, Ph.D

Reviewer 2 Report
The manuscript under the title: “Effect of ambient plasma treatments on thermal conductivity and fracture toughness of boron nitride nanosheets/epoxy nanocomposites” is in line with Nanomaterials journal. This topic is relevant and will be of interest to the readers of the journal. It based on original research. This research has scientific novelty and practical significance. The article has a typical organization for research articles.
Before the publication it requires significant improvements, especially:
1. 1. The "Introduction" section: The functionalization of the surface of nanofillers is very effective, however, it is carried out both by chemical methods and by treating nanomaterials with various physical fields. Consider various options for the functionalization of nanoparticles. I think the related references should be cited corresponding to each aspect, e.g. (but not limited to these), which will undoubtedly improve the "Introduction" section:
- Russ J Appl Chem 86 , 765–771 (2013). https://doi.org/10.1134/S107042721305025X
- Polymer Composites, 36, 1891-1898 (2015). https://doi.org/10.1002/pc.23097
- Polymers 2022, 14(21), 4594; https://doi.org/10.3390/polym14214594
- Appl. Polym. Sci. 2019, 136, 47410, https://doi.org/10.1002/app.47410
- Polymer Composites, 39, E2552-E2561 (2018). https://doi.org/10.1002/pc.24832
- Section 2.1. It is necessary to add the physicochemical characteristics of components - give a table with the main physicochemical and technological properties of epoxy resin, hardener and BNNS.
- Error bar should be added in Figure 4(a,b), 5(a,b), 7 (a).
- Why didn't they change the frequency and power during plasma treatment?
- Does the nanoparticles heat up as a result of plasma treatment, and if so, to what temperature?
- How do you explain that the data in Fig. 3(d) do not agree with the data given in Section 2.1., according to which the particle size of the initial BNNS should be 150 nm?
- Fig.4(c). Why was distilled water used in determining the contact angle, and not an epoxy composition?
- Line 258. Read the sentence carefully, it is obvious that there is a typo.
- Line 264-267. It would be good in this case to cite not your previous work, but the work of other scientific groups, especially since work [56] deals with fiber-reinforced composites, and therefore the strengthening mechanism will be completely different. I recommend you cite the following work: Polymers 2020, 12(7), 1437; https://doi.org/10.3390/polym12071437
Author Response
Response to the Referee’s Comments
|
“Effect of ambient plasma treatments on thermal conductivity and fracture toughness of boron nitride nanosheets/epoxy nanocomposites”
The general revisions of the manuscript are noted in the manuscript under the reviewer’s direction.
[Reviewer #2]
- Note of Revised Version:
- The "Introduction" section: The functionalization of the surface of nanofillers is very effective, however, it is carried out both by chemical methods and by treating nanomaterials with various physical fields. Consider various options for the functionalization of nanoparticles. I think the related references should be cited corresponding to each aspect, e.g. (but not limited to these), which will undoubtedly improve the "Introduction" section:
- Russ J Appl Chem 86, 765–771 (2013). https://doi.org/10.1134/S107042721305025X
- Polymer Composites, 36, 1891-1898 (2015). https://doi.org/10.1002/pc.23097
- Polymers 2022, 14(21), 4594; https://doi.org/10.3390/polym14214594
- Appl. Polym. Sci. 2019, 136, 47410, https://doi.org/10.1002/app.47410
- Polymer Composites, 39, E2552-E2561 (2018). https://doi.org/10.1002/pc.24832
à Thanks for your valuable reviews for enhancing the quality of our paper. The recommended papers are in good quality. The other five papers are cited in our manuscript. Please check them.
<Revised or added paragraph and figure>
Introduction: Page 2, Line 64-66;
Hence, further research in this direction is expected to provide more rational guidance and fundamental understanding towards the realization of the theoretical limits of interfacial properties [24-28].
[24] Burmistrov, I.; Mostovoi, A.; Shatrova, N.; Panova, L.; Kuznetsov, D.; Gorokhovskii, A.; Il’inykh, I. Influence of surface modification of potassium polytitanates on the mechanical properties of polymer composites thereof. Russian Journal of Applied Chemistry 2013, 86, 765-771.
[25] Hameed, A.; Islam, M.; Ahmad, I.; Mahmood, N.; Saeed, S.; Javed, H. Thermal and mechanical properties of carbon nanotube/epoxy nanocomposites reinforced with pristine and functionalized multiwalled carbon nanotubes. Polymer Composites 2015, 36, 1891-1898.
[26] Shcherbakov, A.; Mostovoy, A.; Bekeshev, A.; Burmistrov, I.; Arzamastsev, S.; Lopukhova, M. Effect of Microwave Irradiation at Different Stages of Manufacturing Unsaturated Polyester Nanocomposite. Polymers 2022, 14, 4594.
[27] Amirbeygi, H.; Khosravi, H.; Tohidlou, E. Reinforcing effects of aminosilane‐functionalized graphene on the tribological and mechanical behaviors of epoxy nanocomposites. Journal of Applied Polymer Science 2019, 136, 47410.
[28] Zhang, Q.; Bai, G.; Xiao, W.; Sui, G.; Yang, X. Effect of amine functionalized MWCNT‐epoxy interfacial interaction on MWCNT dispersion and mechanical properties of epoxy‐amine composites. Polymer Composites 2018, 39, E2552-E2561.
- Section 2.1. It is necessary to add the physicochemical characteristics of components - give a table with the main physicochemical and technological properties of epoxy resin, hardener and BNNS.
à We really appreciated that you took time to review my paper and gave us the valuable comments towards enhancing the quality of this manuscript. We have carefully examined and revised the manuscript in a thorough manner according to the reviewer’s suggestion. Thank you for your thoughtful comment.
<Revised or added paragraph and figure>
Materials: Page 2, Line 80-83;
The diglycidyl ether of bisphenol-A (DGEBA; YD-128; Kukdo Chemical Co., Korea) was used as the epoxy resin. The epoxide equivalent weight of DGEBA was 185-190 g/equiv., and its density was 1.2 g∙cm-3 at 25 °C. Furthermore, 4,4'-diaminodiphenylmethane (DDM, 45–49 g/equiv.; Sigma-Aldrich Co., USA) was used as the curing agent [29].
Fig. 1. Chemical structures of DGEBA and DDM.
[29] Jin, F. L., Li, X., & Park, S. J. (2015). Synthesis and application of epoxy resins: A review. Journal of Industrial and Engineering Chemistry, 29, 1-11.
- Error bar should be added in Figure 4(a,b), 5(a,b), 7 (a).
à Thanks for your very professional advice. Added error bar to Fig. 4(a,b), 5(a,b), and 7(a). Thanks for your precious time and valuable comments again.
<Revised or added paragraph and figure>
Result and Discussion: Page 6, Line 218; Page 7, Line 247; Page 9, Line 277;
Fig 4(a,b)
Fig 5(a,b)
Fig 7(a)
- Why didn't they change the frequency and power during plasma treatment?
à We really appreciated that you took time to review my paper and gave us the valuable questions towards enhancing the quality of this manuscript. The earlier studies related to polymer-based composites are primarily experimental which lacks sufficient physical understanding about the interfacial interactions and compatibility of nanomaterials with polymer matrix particularly in terms of surface properties. Thus, the surface treatment of nanomaterials is another critical factor which also controls the interfacial properties of nanocomposites. As a result, both experimental as well as theoretical studies of BNSSs as a function of surface treatment time are emphasized in order to achieve optimum benefits. The chemical and morphological properties are performed to evaluate the changes in surface functional group and dispersion states arising from the plasma treatment, confirming the efficiency of the processes. In addition, the practical implications of the surface functional groups for enhancing the interfacial properties (surface free energy) in epoxy-based nanocomposite has been discussed. Thanks for your kind review.
- Does the nanoparticles heat up as a result of plasma treatment, and if so, to what temperature?
à Thanks for your questions. The plasma treatment was processed under room temperature condition (carried out at 28 ℃ for 10 min). Therefore, nanoparticles were not heated up over 28 ℃. Thanks for your precious time and valuable comments again.
- How do you explain that the data in Fig. 3(d) do not agree with the data given in Section 2.1., according to which the particle size of the initial BNNS should be 150 nm?
à Thanks for your valuable time and contribution to our work. A major problem associated with the use of BNSS in epoxy-based nanocomposite is their high hydrophobicity and extremely low dispersibility, which can lead to irreversible agglomeration in a short period of time due to van der Waals interactions between the prismatic surfaces. In, addition, we have carefully examined and revised the manuscript in a thorough manner according to the reviewer’s suggestion. Thanks for your kind review.
<Revised or added paragraph and figure>
Results and Discussion: Page 5, Line 171;
The colloidal stabilities and statistical particle size distributions of the bare Pristine BNNS, PBNNS-7, and PBNNS-10 in distilled water were examined via sedimentation tests and dynamic laser scattering (DLS) measurements performed 24 h after ultra-sonication for 30 min. In this technique, the random changes of 2D structure materials in the liquid medium are analyzed according to the intensity of scattered light and compared with that of an equivalent hard sphere of ceramic materials. This can provide an estimate of the apparent size of the nanosheet agglomerates in the suspension. A stable colloid will have a constant mean particle size over a period of time, whereas an unstable colloid will exhibit an increase in particle size over time.
- Fig.4(c). Why was distilled water used in determining the contact angle, and not an epoxy composition?
à Thanks for your valuable time and contribution to our work. Contact angle measurement is one of the most suitable instruments for determining the surface free energy of epoxy nanocomposites. There are three wetting liquids what distilled water, ethylene glycol, and diiodomethane for contact angle measurement. It has been widely used in various professional studies [1], [2], [3], [4]. Thanks for your kind review.
[1] Turunen, M. P., Laurila, T., & Kivilahti, J. K. (2002). Evaluation of the surface free energy of spin‐coated photodefinable epoxy. Journal of Polymer Science Part B: Polymer Physics, 40(18), 2137-2149.
[2] Chibowski, E. (2003). Surface free energy of a solid from contact angle hysteresis. Advances in colloid and interface science, 103(2), 149-172.
[3] Hejda, F., Solar, P., & Kousal, J. (2010, June). Surface free energy determination by contact angle measurements–a comparison of various approaches. In WDS (Vol. 10, pp. 25-30).
[4] Bahmani, H., Sanij, H. K., & Peiravian, F. (2021). Estimating Moisture Resistance of asphalt mixture containing epoxy resin using Surface Free Energy Method and Modified Lottman test. International Journal of Pavement Engineering, 1-13.
Table. Surface free energy, specific of the test wetting liquids used.
|
Wetting liquids |
(mJ.m-2) |
(mJ.m-2) |
(mJ.m-2) |
|
Distilled water |
72.80 |
21.80 |
0.38 |
|
Ethylene glycol |
47.70 |
31.00 |
16.70 |
|
Diiodomethane |
50.80 |
50.42 |
51.00 |
- Line 258. Read the sentence carefully, it is obvious that there is a typo.
à Thanks for your valuable reviews for enhancing the quality of our paper. We improved the English sentences. Thanks for your revise.
Introduction: Page 1, Line 26-27;
Boron nitrides (BNs) are identified as an intriguing candidate for fillers in electrically insulating polymer-based nanocomposites due to their relatively high thermal conductivity and good mechanical properties.
Introduction: Page 9, Line 287-288;
The highest measured thermal conductivity value is 21.4 MPa.m1/2 for the PBE-7 nanocomposite, indicating the 110% enhancement compared to the neat BNNS (10.7 MPa.m1/2).
- Line 264-267. It would be good in this case to cite not your previous work, but the work of other scientific groups, especially since work [56] deals with fiber-reinforced composites, and therefore the strengthening mechanism will be completely different. I recommend you cite the following work: Polymers 2020, 12(7), 1437; https://doi.org/10.3390/polym12071437
à Thanks for your valuable reviews for enhancing the quality of our paper. The recommended papers are in good quality. Another paper is cited in our manuscript. Thanks for your revise.
Result and discussion: Page 9, Line 293-296;
This means that any cracks formed will propagate through the epoxy matrix and around the PBNNS due to the outstanding dispersion and interfacial adhesion between the PBNNS and the epoxy matrix, thus leading to good fracture and pull-out resistance [68,69].
[68] Bekeshev, A., Mostovoy, A., Tastanova, L., Kadykova, Y., Kalganova, S., & Lopukhova, M. (2020). Reinforcement of epoxy composites with application of finely-ground ochre and electrophysical method of the composition modification. Polymers, 12(7), 1437.
We really appreciated that you took time to review my paper and gave us the valuable comments towards enhancing the quality of this manuscript.
Sincerely yours,
Soo-Jin Park, Ph.D

Round 2
Reviewer 2 Report
The authors considered most of the comments or adequately responded to the remarks contained in the review; therefore, the work may be approved for publication.